# Inkjet-Printed Phospholipid Bilayers on Titanium Oxide Surfaces: Towards Functional Membrane Biointerfaces

**DOI:** 10.3390/membranes12040361

**Published:** 2022-03-25

**Authors:** Sigalit Meker, Oded Halevi, Hokyun Chin, Tun Naw Sut, Joshua A. Jackman, Ee-Lin Tan, Michael G. Potroz, Nam-Joon Cho

**Affiliations:** 1School of Materials Science and Engineering, Nanyang Technological University, 50 Nanyang Drive, Singapore 637553, Singapore; sigi.meker@gmail.com (S.M.); full.ahead.hogyun@gmail.com (H.C.); eelintan05@gmail.com (E.-L.T.); mikepotroz@gmail.com (M.G.P.); 2The Institute of Chemistry, The Hebrew University of Jerusalem, Jerusalem 91904, Israel; odedqs@gmail.com; 3School of Chemical Engineering and Translational Nanobioscience Research Center, Sungkyunkwan University, Suwon 16419, Korea; suttunnaw@skku.edu (T.N.S.); jjackman@skku.edu (J.A.J.)

**Keywords:** lipid bilayer, inkjet printing, titanium dioxide, bioinspired coating, micropatterning

## Abstract

Functional biointerfaces hold broad significance for designing cell-responsive medical implants and sensor devices. Solid-supported phospholipid bilayers are a promising class of biological materials to build bioinspired thin-film coatings, as they can facilitate interactions with cell membranes. However, it remains challenging to fabricate lipid bilayers on medically relevant materials such as titanium oxide surfaces. There are also limitations in existing bilayer printing capabilities since most approaches are restricted to either deposition alone or to fixed microarray patterning. By combining advances in lipid surface chemistry and on-demand inkjet printing, we demonstrate the direct deposition and patterning of covalently tethered lipid bilayer membranes on titanium oxide surfaces, in ambient conditions and without any surface pretreatment process. The deposition conditions were evaluated by quartz crystal microbalance-dissipation (QCM-D) measurements, with corresponding resonance frequency (Δf) and energy dissipation (ΔD) shifts of around −25 Hz and <1 × 10^−6^, respectively, that indicated successful bilayer printing. The resulting printed phospholipid bilayers are stable in air and do not collapse following dehydration; through rehydration, the bilayers regain their functional properties, such as lateral mobility (>1 µm^2^/s diffusion coefficient), according to fluorescence recovery after photobleaching (FRAP) measurements. By taking advantage of the lipid bilayer patterned architectures and the unique features of titanium oxide’s photoactivity, we further show how patterned cell culture arrays can be fabricated. Looking forward, this work presents new capabilities to achieve stable lipid bilayer patterns that can potentially be translated into implantable biomedical devices.

## 1. Introduction

Implantable devices have become exceedingly sophisticated in recent years [1] and the need for device biocompatibility has led to purpose-driven development of bioinspired interfaces [2,3,4,5,6]. Bioinspired thin-film coatings can improve the integration of artificial devices within the body and can enable communication with the surrounding tissue environment. Such responsive functional features of the bioinspired coating may include antifouling, immunocloaking, and cell-activating properties [2]. A natural communicating and responsive interface in the cellular environment is the phospholipid bilayer [7,8,9]. This makes the lipid bilayer a principal component of advanced biointerfaces, giving rise to the concept of solid-supported lipid bilayers as thin-film coatings. 

Nevertheless, solid-supported lipid membrane fabrication is highly challenging for several reasons: (1) Preserving the nanostructured architecture of phospholipid membrane is vital for membrane functionality [10]. As such, lipid bilayer fabrication requires the vesicle fusion technique, which is limited to mainly silica-based materials [11]. (2) The intermolecular interactions that hold together the lipid membrane are relatively weak, noncovalent forces, and therefore often have limited utility as durable coating interfaces. Additionally, the lack of strong interactions between the implant surface and phospholipid molecules typically leads to bilayer spreading and to high air sensitivity [12]. (3) Implantable devices are often made of metal and metal oxides, such as titanium-based implants covered with a thin TiO_2_ layer [13,14,15]. Vesicle fusion is conventionally ineffective on TiO_2_ surfaces due to weak interactions with phosphatidylcholine (PC) lipid vesicles and other common lipids [16]. (4) For optimal utilization of lipid membranes as functional coating materials, a rapid deposition method is required to minimize the material consumption, and to provide precise control over the location of the deposited materials, with potential translation to three-dimensional objects. 

Considering these points, the inkjet printing method is an excellent candidate for lipid bilayer coating production [17,18,19]. However, utilizing a rapid deposition method for the formation of lipid membranes on a flat surface is challenging due to competing design needs. Recently, pioneering work describing inkjet printing of phospholipid bilayers on a glass substrate was performed by lipid deposition into a pre-patterned confined area that was formed by photolithography [20]. Consequently, the inkjet printer was not employed as a selective patterning tool of lipid bilayers, but rather mainly as a deposition tool. In a different work, an array of lipid bilayers was spotted on a glass substrate [21], however no specific patterns could be formed and constant wetting of the substrate during and after deposition was required to prevent dehydration that ruined the bilayer. These challenges highlight the need to explore more diverse lipid chemistry options that open up new functionalization opportunities for printing applications, especially ones that enable simultaneous deposition and patterning. 

Towards this goal, the synthetic phospholipid 2-((2,3-bis (oleoyloxy) propyl) dimethylammonio) ethyl hydrogen phosphate (DOCP) has been introduced to functionalize oxide nanoparticles [22,23,24]. It is an inverted headgroup derivative of the naturally occurring 1,2-dioleoyl-*sn*-glycero-3-phosphocholine (DOPC) lipid, which is one of the most common phospholipids in natural cell membranes (Figure 1a) [22,23]. DOCP has an exposed phosphate group that interacts strongly with TiO_2_ via coordinate binding to Ti atoms [23,24,25,26,27,28]. This coupling is commonly used to anchor molecules, polymers, and dense monolayers to oxide surfaces [25,26,28]. Notably, DOCP lipid vesicles were reported to rupture and form stable lipid bilayers on TiO_2_ nanoparticles [24,29]. Due to DOCP lipid chemisorption on the surface, the resulting bilayer coatings were shown to be resistant to extreme pH, hydrogen-bond breaker, and surfactant treatments, as well as to exposure to albumin protein and serum solutions [24]. More recently, it was also shown that DOCP lipid vesicles can fuse with and rupture on planar TiO_2_ surfaces to form supported lipid bilayer coatings, whereby the lower lipid leaflet is covalently attached to the substrate [30]. These features make DOCP lipid bilayers particularly durable and thus an excellent candidate for utilization as a selective biocompatible coating and functional biointerface for titanium-based devices, especially by taking advantage of robust and rapid deposition methods such as inkjet printing to fabricate rugged lipid bilayer coatings with high precision and on demand. Herein, this work is directed at developing microprinting strategies for fabrication of DOCP phospholipid bilayer coatings on TiO_2_ surfaces and utilizing patterned membrane nanostructures to regulate the spatial dynamics of cell–substrate interactions. 

## 2. Materials and Methods

*Materials:* 2-((2,3-bis (oleoyloxy) propyl) dimethylammonio)ethyl hydrogen phosphate (DOCP), 1,2-dioleoyl-*sn*-glycero-3-phospho-L-serine (sodium salt) (DOPS), 1,2-dioleoyl-*sn*-glycero-3-phosphoethanolamine-N-(lissamine rhodamine B sulfonyl) (ammonium salt) (Rhod-PE), and 1-oleoyl-2-{12-[(7-nitro-2-1,3-benzoxadiazol-4-yl) amino] dodecanoyl}-sn-glycero-3-phosphate (NBD-PA) lipids were purchased from Avanti Polar Lipids (Alabaster, AL, USA). The Annexin V-FITC Apoptosis Kit for the annexin-5A binding study was purchased from BioVision Inc (Waltham, MA, USA). Clear borosilicate glass coverslips, covered with a thin TiO_2_ layer, ~15 nm, were obtained from the Deposition Research Laboratory, Inc (St. Charles, MO, USA).

*Materials for SEM:* 8 mm × 20 m adhesive carbon tape was obtained from Agar Scientific Ltd. (Stansted, Essex, United Kingdom), 10 mm × 30 m adhesive copper tape was purchased from EMSIS ASIA Pte Ltd. (Singapore), and 25 × 16 mm angled (4°) aluminum SEM specimen stub and 25 × 16 mm aluminum SEM specimen stub were both purchased from EMSIS ASIA Pte Ltd.

*Materials for Cell Culture:* MEM Alpha Modification, with L-Glutamine, with Ribo- and Deoxyribonucleosides (MEM), was purchased from HyClone, GE Healthcare Life Science (Boston, MA, USA). Fetal bovine serum, antibiotic-antimycotic, and the LIVE/DEAD Viability/Cytotoxicity Kit for cell staining were purchased from Thermo Scientific Pte Ltd. (Waltham, MA, USA). Human lung fibroblast cells, MRC-5, were obtained from RIKEN Bio Resource Center (BRC, Kyoto, Japan).

*Vesicle Preparation:* Small unilamellar vesicles were obtained by the extrusion method, as previously reported [31]. Lipids were purchased as chloroform stock solutions. The lipids were dried under a gentle stream of N_2_ to achieve thin films, and left to dry overnight under vacuum to remove any residual organic solvents. Before extrusion, the lipid film was hydrated in Tris buffer, 10 mM, pH 7.5, 150 mM NaCl, to form multilamellar vesicles. Vesicles in buffer, 5 mg/mL, were vortexed and the resulting multilamellar vesicles were extruded by using a Mini Extruder (Avanti Polar Lipids, Alabaster, AL) through a polycarbonate membrane of 50 nm-diameter pores. Vesicle solutions were diluted before the experiment with the same buffer to the desired concentration, and were used fresh within 3 days of preparation. All buffers and solutions were prepared with 18.2 MΩ·cm Milli-Q-treated water (MilliporeSigma, Burlington, MA, USA).

*Dynamic Light Scattering:* Vesicle size was assessed by a 90 Plus Particle Size Analyzer (Brookhaven Instruments Corporation, New York, NY, USA) at a scattering angle of 90°, where the reflection effect is minimized. All vesicles exhibited size distribution of 70–80 nm with polydispersity of <0.1.

*Quartz Crystal Microbalance-Dissipation (QCM-D):* The experiments were conducted on a Q-Sense E4 instrument (Q-Sense AB, Gothenburg, Sweden). Experimental data were collected at several overtones (*n* = 3, 5, 7, 9), and the changes in frequency (Δƒ) and energy dissipation (ΔD) were monitored as a function of time [32]. The reported values are from the seventh overtone (*n* = 7) and were calculated as an average with standard deviation from 3 repeats. All measurements were performed on QCM-D sensor crystals (Q-Sense AB) with ~50 nm-thick, titanium oxide coatings. The crystals were washed with 1% *w*/*v* sodium dodecyl sulfate (SDS) solution, followed by rinsing with water and ethanol. The crystals were subjected to oxygen plasma treatment (Harrick Plasma, Ithaca, NY, USA) for ∼4 min immediately before the experiments. 

*FRAP Mobility Study:* The fluorescence recovery after photobleaching (FRAP) technique was employed to measure the lipid bilayer diffusivity on the TiO_2_ substrate. Fluorescence images were captured with an inverted epifluorescence Eclipse TE 2000 microscope (Nikon, Tokyo, Japan) equipped with a 60× oil immersion objective (NA 1.49) and an Andor iXon + EMCCD camera (Andor Technology, Belfast, Northern Ireland). The samples were measured by a TRITC (Rhodamine-DHPE) filter set with a mercury lamp (Intensilight C-HGFIE; Nikon Corporation, Tokyo, Japan). The size of the bleaching spot was around 25 µm in diameter, and the bleaching process was performed with a 532 nm, 100 mW power laser beam. The recovery process was recorded from −5 to 120 s at 2 s intervals to monitor the fluorescence intensity signal. FRAP analysis was performed by Matlab software to calculate the diffusion coefficients based on the Hankel transform method [33]. 

*Patterned Bilayers by Inkjet Printing:* The ink was a 1 mg/mL dispersion of DOCP unilamellar vesicles, prepared as detailed above, with 1% w/w of Rhod-PE or NBD-PA, in Tris 10 mM, 150 mM NaCl, pH 7.5 buffer. No additives were added to the ink. The substrate was coverslip glass-sputtered with a titanium dioxide layer, and was used as purchased. Inkjet printing of the desired patterns was performed using an Omnijet100 printer (UniJet, Hwaseong, Korea), with a Dimatix 10 picoliter printing head. The substrate temperature during printing was 70 °C. The ink cartridge temperature was set to room temperature. Printing was conducted at 800 Hz and 1000 dpi. The waveform consisted of a 3 μs rise time to 22 volts, 4 μs rise time to 27 volts, and another 3 μs fall time to 0 volts. The printed pattern was left at room temperature overnight to allow maximal binding of the lipids and the substrate. Following this, the sample was washed with Milli-Q-treated water to remove the excess lipids. Fluorescence images were taken with a Leica DMI6000B microscope (Leica, Wetzlar, Germany) and a Zeiss 710 Confocal Microscope (Carl Zeiss AG, Jena, Germany).

*Confocal Laser Scanning Microscopy (CLSM) Analysis:* CLSM was performed using a Zeiss 710 Confocal Microscope (Carl Zeiss AG, Germany). Rhodamine B sulfonyl-conjugated lipids, Rhod-PE, were used in the preparation of lipid bilayers. Cell samples were incubated (1 h, 37 °C, 5% CO_2_) with calcein acetoxymethyl in culture media (1 μL/mL) prior to their imaging. Characterization of bilayers and cells was performed with samples in cell culture media in cell culture plates. Imaging was performed successively with three laser excitation channels: 405, 488, and 561 nm, with three respective emission filters: 416–477, 498–550, and 572–620 nm, and objective lenses (20× or 10×). Z-stack slices were collected for each sample. Images were processed for brightness and contrast using ImageJ.

*Scanning Electron Microscopy (SEM):* Samples, i.e., DOCP bilayer prints on TiO_2_-coated glass slides, were mounted on a flat or angled specimen stub using carbon and copper adhesive tapes. Samples were not coated before characterization. Samples were imaged using FESEM JSM-7600F (JEOL Ltd., Tokyo, Japan) at 80× and 1000× magnifications, with an acceleration voltage of 1.50 and 1.0 kV, respectively. For cell studies, cells were fixed with 4% glutaraldehyde for 30 min, washed three times with PBS, followed by gentle freezing in liquid nitrogen and freeze-drying for 24 h before characterization.

*Cell Adhesion Study:* MRC-5, human lung fibroblasts cells, were cultured in MEM media with fetal bovine serum (10%) and antibiotic-antimycotic (1%). Printed glass slides were attached to the bottom of a petri dish. Cells were then seeded, at a 0.2–0.5 × 10^6^ density, in the aforementioned petri dish with culture media. Cells were incubated overnight for 16 h, at 37 °C, 5% CO_2_, to enable cell adhesion and to reach ~70–80% confluency, following which the samples were washed three times with fresh media to remove any unattached cells. Cell viability was monitored with an optical microscope throughout the experiment and was characterized with CLSM at the end of the incubation. Cells were stained with calcein acetoxymethyl (Live/Dead assay) prior to CLSM characterization, as described above.

## 3. Results and Discussion

We first conducted QCM-D measurements to track DOCP lipid interactions with planar TiO_2_ surfaces. Small unilamellar vesicles (SUVs) of 100 mol% DOCP phospholipid were extruded to afford monodispersed vesicles in 10 mM Tris buffer (pH 7.5) with 150 mM NaCl [31]. These conditions favor the formation of phospholipid bilayers on oxide surfaces [34]. DOCP lipid vesicles were injected into the QCM-D measurement chamber at 24 °C, and their interactions with the TiO_2_ surface were monitored by recording shifts in resonance frequency (Δf, Hz) and energy dissipation (ΔD, 10^−6^) (Figure 1b). The Δf shifts correlate with an increase in adsorbed mass on the sensor surface, whereas an increase in energy dissipation relates to higher viscoelasticity of the adlayer [34,35]. The measured frequency and energy dissipation shifts were typical of phospholipid bilayer formation, with the following values [32,33]: Δf_n=3_ = −25.5, Δf_n=5_ = −25.4, Δf_n=7_ = −25.5 Hz, and ΔD_n=3_ = 0.09, ΔD_n=5_ = 0.1, ΔD_n=7_ = 0.07 × 10^−6^. This result agrees with previous reports relating to the spontaneous formation of a DOCP lipid bilayer coating on TiO_2_ nanoparticles and planar surfaces under similar solution conditions [24,29,30].

As mentioned above, forming planar phospholipid bilayers on TiO_2_ surfaces is challenging using conventional lipids and limited either to highly specific lipid compositions or to particular buffer/solvent conditions, or requires the catalytic addition of membrane-active peptides [34,36,37,38,39,40]. For comparison, we monitored the interactions of zwitterionic DOPC lipid vesicles with a TiO_2_ surface under equivalent solution conditions. As expected, DOPC lipid vesicles did not rupture on the TiO_2_ surface and instead remained as adsorbed intact vesicles (Figure 1c): Δf_n=3_ = −167.6, Δf_n=5_ = −156.8, Δf_n=7_ = −147.8 Hz, and ΔD_n=3_ = 16.6, ΔD_n=5_ = 14.7, ΔD_n=7_ = 12.0 × 10^−6^. The frequency and energy dissipation shifts agree with previous reports and confirm that adsorbed DOPC lipid vesicles remained intact on the TiO_2_ surface [16,34,35,41]. These differences between DOPC and DOCP lipid vesicle interactions with TiO_2_ are an outcome of the chemisorption of DOCP lipid vesicles onto TiO_2_ [24,26,27,28,29]. Control experiments, in which DOCP lipid bilayers were washed with ethanol, verified that following the ethanol wash, the lower leaflet, which was tethered to the surface, remained intact [30] (Appendix A). This finding verified that DOCP lipids were indeed bound to the TiO_2_ surface, otherwise both leaflets would have been washed away with ethanol. On the other hand, DOPC, which has a hindered phosphate group, did not form such chemical bonds, and hence DOPC lipid vesicles did not rupture [23,24]. Overall, DOCP lipid vesicles, in contrast to conventional DOPC lipid vesicles, formed a covalently tethered phospholipid bilayer coating on the TiO_2_ surface, which provides a versatile chemistry for inkjet printing that is driven by spontaneous interactions and molecular self-assembly.

The strong interactions of DOCP lipid vesicles with TiO_2_ surfaces facilitated the formation of tethered lipid bilayers, which can be employed for coating TiO_2_ surfaces. While a conformal coating is a promising first step, a patterned coating can be particularly useful for many applications. For example, a selective coating comprising micropatterned lipid bilayers can form well-defined areas that are accessible to cells, while other regions repel cells. Overall, selective deposition of the lipid bilayer coating provides more options and flexibility for the design of device interfaces. For this aim, we utilized inkjet printing to form micropatterns of DOCP tethered bilayers on a bare, untreated TiO_2_ surface (Figure 1d). The ink composition was simply DOCP SUVs, which were extruded [31] to afford monodispersed vesicles in 10 mM Tris buffer (pH 7.5) with 150 mM NaCl. 

In the printing process, one of the most challenging factors to control is the evaporation rate of the printed dispersion. On one hand, it should be slow enough to allow covalent DOCP lipid binding to the TiO_2_ surface, vesicle rupture, and bilayer self-assembly before dehydration. On the other hand, the rate should be fast enough to form well-defined and consistent patterns without extensive spreading across the hydrophilic surface. Two important parameters governing the net evaporation rate were the substrate temperature and the jetting frequency of the ink drops from the nozzles (Figure 2). 

The jetting frequency had a relatively minor effect on the printing quality, and similar results were achieved across a range of frequencies (200–1000 Hz). Regarding substrate temperature, the QCM-D results showed that DOCP lipid binding to the TiO_2_ surface occurred rapidly at 24 °C (cf. Figure 1b). However, during printing under this temperature condition, the ink spread across the substrate and produced low-resolution and non-uniform patterns (Figure 2a). This issue was caused by the low evaporation rate of the solvent and the extensive wetting of the substrate by the ink. To address this challenge, the substrate temperature was increased to 70 °C and solvent evaporation occurred quickly enough to enable the formation of higher-resolution patterns, with lines of about 40 µm in width (Figure 2b). Combined with a jetting frequency of 800 Hz, uniformed and defined patterns of DOCP tethered bilayers were printed (Figure 2c). All printing steps were conducted with one printed layer only.

Following printing, the dried lipid patterns were covered with excess lipids and salt residues from the evaporated buffer solution, as revealed in scanning electron microscopy (SEM) images (Figure 3). The dried lipid patterns were stable under ambient conditions for at least several weeks before further characterization. By gently rinsing the lipid patterns with water, the salt residues were easily removed and the bilayer’s micropatterned structures were preserved (Figure 3). SEM experiments showed that the lipid patterns had tunable widths and lengths depending on the printed pattern, and a minimal typical width of 40 µm.

Following hydration of the lipid patterns in buffer solution, lipid bilayer alignment in the patterns was evaluated by measuring lateral lipid diffusivity with the fluorescence recovery after photobleaching (FRAP) technique [33]. The diffusivity of printed lipid bilayers was assessed and compared with that of non-patterned DOCP lipid bilayers, which were formed via vesicle fusion across the entire TiO_2_ surface (Figure 4). Notably, FRAP measurements revealed similar diffusivity of the lipids in the printed and non-printed lipid bilayers, with values of 1.2 ± 0.3 and 1.3 ± 0.1 µm^2^/s, respectively (Figure 4a,b,d). In contrast, adsorbed layers of intact DOPC lipid vesicles on TiO_2_ were immobile (Figure 4c,d). The results indicate that printed DOCP lipid bilayers were likely aligned in a two-leaflet structure with characteristic membrane fluidity. Indeed, while the diffusivity coefficients values were approximately two-fold lower compared to noncovalently attached planar DOPC lipid bilayers formed on SiO_2_ [42], the reduced fluidity is likely an outcome of the tethered nature of the lower leaflet. Overall, the FRAP measurements indicate that the inkjet-printed bilayers constituted functional membranes. Moreover, as the FRAP experiments were conducted after the rehydration of the dry printed lipid bilayers, the results demonstrate their stability to air and dehydration, and hence high durability.

We further investigated the potential of micropatterned lipid bilayers to corral adhering cells. To evaluate cellular behavior in the vicinity of printed DOCP lipid bilayers, the printed bilayers were incubated overnight with MRC-5 cells, a human lung fibroblast non-cancerous cell line, in the presence of 10% serum (Figure 5). Cells attached to and covered the bare TiO_2_ surface, reaching around ~80% confluency, but did not attach to the printed lipid bilayers. These observations agree with previous reports that in the presence of serum proteins, mammalian cells favor attachment to TiO_2_ over neutral phospholipid bilayers [43,44,45]. This effect occurs because phospholipid bilayers by themselves exhibit intrinsic antifouling effects, which prevent protein adsorption and extracellular matrix (ECM) deposition, and hence limit cell adhesion. 

The interface between the patterned lipid bilayer and the adhering cell layer was characterized by confocal laser scanning microscopy (CLSM) and SEM (Figure 5a–f). Following overnight incubation for 16 h, the samples were washed to remove unattached cells. For CLSM visualization, cells were labeled with calcein (green channel) and lipids with 1% dye-conjugated lipid, 1,2-dioleoyl-*sn*-glycero-3-phosphoethanolamine-N-(lissamine rhodamine B sulfonyl) (DOPE-Rhod) (red channel). For SEM characterization, cells were first fixed with glutaraldehyde and then freeze-dried. Cells were observed under the optical and fluorescence microscopes and appeared viable and without any visible signs of stress under the applied experimental conditions. Notably, the cells remained attached adjacent to the prints (Figure 5b,d), suggesting that the prints did not affect cell viability.

Interestingly, a polymer carrying conjugated CP groups, similar to the DOCP headgroup, was reported to promote cell adhesion [46], while our findings indicate that the printed bilayers hinder cell adhesion directly on the prints and demonstrate antifouling characteristics. This finding is consistent with the antifouling character of neutral lipid bilayers [43,44,45]. A possible clarification is found when considering the degree of freedom of the CP headgroup. The flexibility and free motion of the polymer-conjugated CP groups were reported to be crucial for attachment to cell membranes and consequently cell adhesion [46]. It is possible that the orientation and alignment of the DOCP lipids in the tightly packed bilayers limit their degree of freedom, and hence they are unable to form strong interactions with cell membranes. Notably, this antifouling nature can be utilized to print confined growth areas for cells on micro- and macroscopic scales depending on the preferred design. 

To demonstrate the ability to control cellular distributions over the substrate, cell patches of 0.08 ± 0.01 mm^2^ area were formed and confined by the printed bilayer region (Figure 5d,e). This feature has the potential to prevent cell growth in specific locations on a device surface, while at the same time forming areas that are approachable to cells and patches that are resistant to cells. Overall, the DOCP phospholipid bilayer prints enabled cell adhesion in controlled regions without any noticeable signs of stress.

TiO_2_ nanoparticles and nano-coated surfaces typically demonstrate photoactivity and induce production of oxygen radicals upon UV light exposure [47]. It is well-established that the attachment of phosphate groups, similar to the DOCP headgroup, can increase TiO_2_ photoactivity and the tendency to form radicals [48,49,50,51,52,53,54,55]. Therefore, in the near vicinity of the printed DOCP lipid bilayers and upon short exposure to a visible laser with three excitation channels, 405, 488, and 561 nm, cells gradually started to detach from the surface (Figure 6). Cell detachment only occurred near the printed bilayers, and not in distinct regions on the surface. This observation supports that cell detachment upon laser exposure is due to bilayer-enhanced photoactivity of the TiO_2_ surface in that particular location, i.e., increased production of oxygen radicals that interfere with cell adhesion and viability [47,48,49,50,51,52,53,54,55]. The responsiveness of the surface to visible laser light may be beneficial when cells have to be detached or removal of an implantable device is required.

## 4. Conclusions

We utilized inkjet printing for the formation of patterned DOCP lipid bilayers on a TiO_2_ surface that can be used as a functional biointerface. As surface pretreatment was not necessary and the process can proceed without requiring any additional reagents, the printer was used as both a deposition and patterning tool, and therefore was not limited to microarray patterning. Compared to a previous report [20] that described inkjet printing of a phospholipid bilayer on a glass surface, there are several compelling merits and novel elements of the surface functionalization pursued here: (1) the current approach involves covalent attachment of the phospholipid bilayer to the TiO_2_ surface, whereas the previously printed bilayer on glass was only attached via noncovalent forces, and (2) no TiO_2_ surface pretreatment was required in the present case, whereas a polymerized lipid bilayer micropattern was first fabricated on the surface in the previous report prior to inkjet printing that resulted in a far more complex process overall. On the other hand, the streamlined fabrication capabilities we have developed in the present case in turn enabled the printed DOCP lipid bilayers on TiO_2_ to exhibit high durability, as dehydration did not damage the printed patterns. Importantly, upon rehydration, the bilayer mobility was regained. This feature makes DOCP lipid bilayers highly useful for biomedical applications by overcoming typical sensitivity to air exposure and limitations of dehydration/rehydration. Pattern features as small as 40 µm were achieved based on the optimized printer settings used in this case, and the inkjet printer from this study can be potentially expanded to areas as large as 500 × 500 mm, especially if future design iterations can bypass the need for preparing lipid vesicles as new coating methods emerge [56,57,58].

From an applied perspective, the combination of DOCP lipid bilayer durability on TiO_2_ surfaces and convenient inkjet printing deposition increases the applicability of the planar bilayer as a bioinspired coating technology, which could be advantageous for titanium-based medical implants that typically have TiO_2_ oxide film thicknesses ranging from around 3 to 1000 nm [59]. In the present case, the film thickness was ~15 to ~50 nm in the different experiments, but the fabrication approach is anticipated to be broadly useful across a wide range of film thickness values because the chemisorption process only depends on coordinate bond formation between DOCP lipid molecules and the uppermost TiO_2_ surface layer. For example, upon exposure of the printed bilayers to cells, the bilayer demonstrated antifouling characteristics when cells attached only to the bare surface, due to corralling by the patterned bilayers. While the pattern design we used was relatively simple in this proof-of-concept study (cf. Figure 5d), more sophisticated patterns could potentially be incorporated to modulate cell–implant interactions by controlling pattern shapes and dimensions. It might be advantageous to use the patterned bilayer coatings to corral cells to interact with implant surfaces in a more controlled manner while modulating cell–cell interactions and potentially improving biocompatibility. Such functionality might be utilized to prevent unwanted cell growth on implants and medical devices at strategic locations, and the bilayer coating itself could also reduce nonspecific adsorption of immune-related proteins. Looking forward, the integration of patterned bilayers with additional signaling ligands, such as glycolipids and lipopolysaccharides, can lead to highly functional, biocompatible microdevices for biomedical applications.

## Figures and Tables

**Figure 1 membranes-12-00361-f001:**
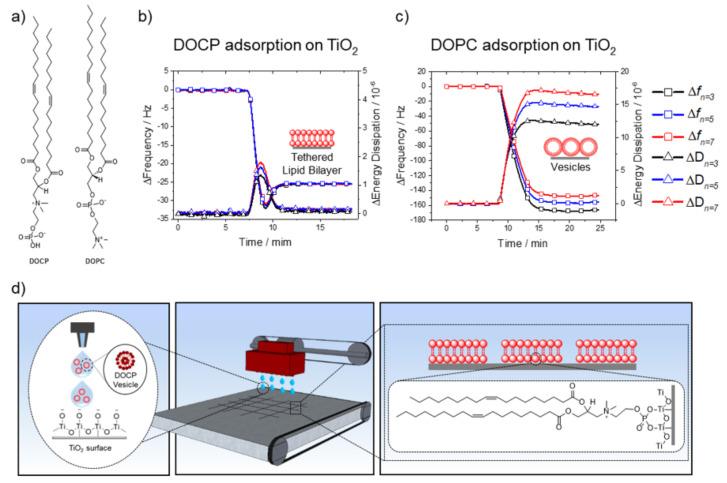
(**a**) Chemical structures of DOCP and DOPC phospholipids: DOCP is an inverted headgroup derivative of the zwitterionic DOPC phospholipid. (**b**) QCM-D plot of DOCP lipid vesicle adsorption onto a TiO_2_ surface, and subsequent rupture to form a planar and tethered lipid bilayer (decreased frequency (Hz) indicates mass increase of the lipid adlayer, and increased energy dissipation (10^−6^) correlates with film viscoelasticity). (**c**) QCM-D plot of DOPC lipid vesicle adsorption onto a TiO_2_ surface, whereby adsorbed vesicles remained intact and did not rupture. (**d**) Schematic illustration of inkjet-printed DOCP lipid vesicles to form macroscale and microscale patterns comprising tethered phospholipid bilayers on a TiO_2_ surface.

**Figure 2 membranes-12-00361-f002:**
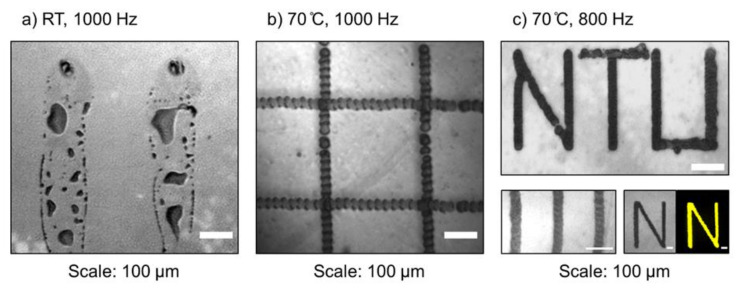
Camera images of printed lipid pattern on TiO_2_ substrates. Effect of substrate temperature and inkjet frequency on printing quality: (**a**) DOCP lipid vesicles printed at room temperature, 1000 Hz. Non-uniform patterns with low resolution. (**b**) DOCP lipid vesicles printed at 70 °C, 1000 Hz. Uniform and high-resolution patterns. (**c**) DOCP lipid vesicles printed at 70 °C, 800 Hz. Uniform patterns were formed. The change in frequency had a minor effect. DOCP pattern imaged with a fluorescence microscope (lower right, in yellow).

**Figure 3 membranes-12-00361-f003:**
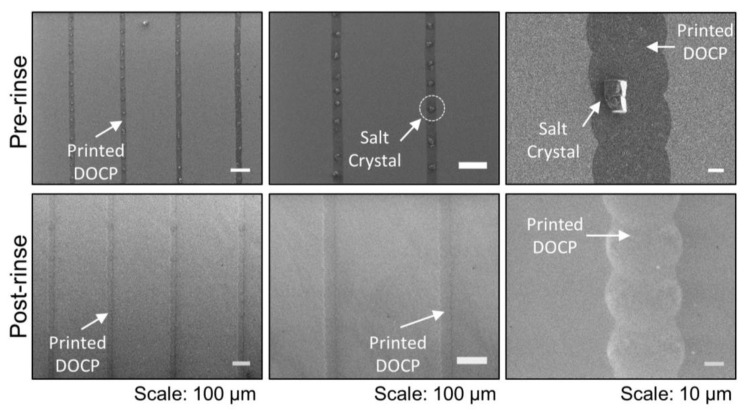
SEM characterization of inkjet-printed tethered phospholipid bilayers on a TiO_2_ surface: top—crude printed bilayers at different magnifications with salt crystals present, which originate from the buffer, immediately after printing, and bottom—printed bilayers following water rinse to remove salt crystal residues.

**Figure 4 membranes-12-00361-f004:**
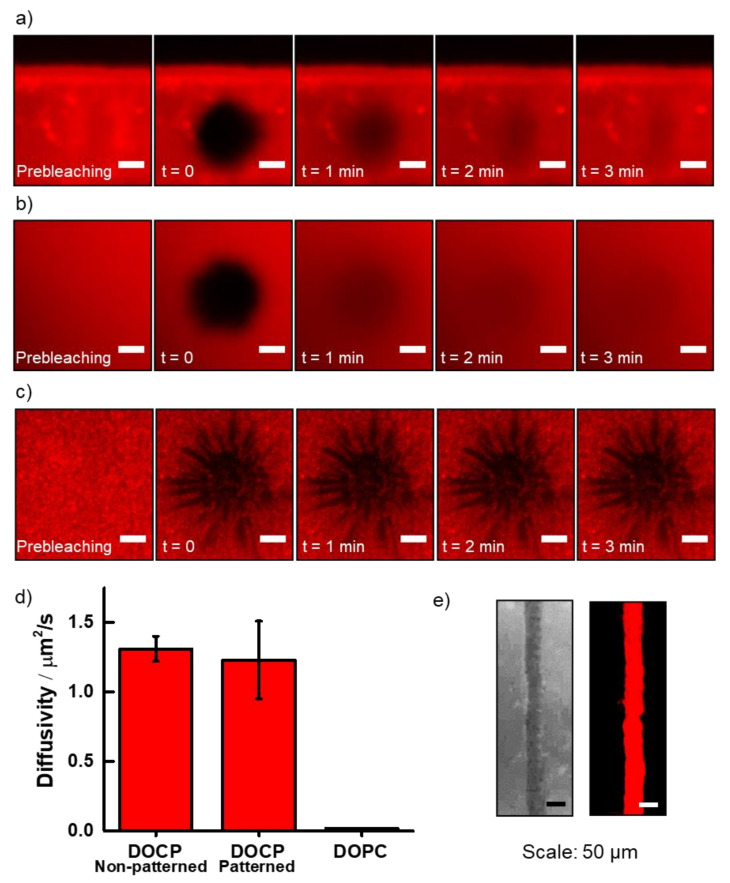
Fluorescence recovery after photobleaching (FRAP) experiments to measure lateral phospholipid diffusion in tethered lipid bilayer pattern on a TiO_2_ surface: (**a**) Photobleaching and time-dependent recovery of inkjet-printed phospholipid bilayers in micropatterns. The fluidity of the DOCP phospholipids confirms that the lipids exhibit a fluidic membrane character. (**b**) Photobleaching and recovery of DOCP phospholipid bilayer prepared by vesicle fusion across a TiO_2_ surface. (**c**) Photobleaching without recovery for a DOPC intact vesicle adlayer on a TiO_2_ surface. FRAP image scale bars: 10 µm. (**d**) Diffusion coefficients of DOCP lipid bilayer fabricated by inkjet printing (**a**) compared with DOCP lipid bilayer fabricated by vesicle fusion (**b**) and DOPC lipid bilayer fabricated by vesicle fusion (**c**). (**e**) DOCP printed line captured in lower magnification with a camera (**left**) and with a fluorescence microscope (**right**).

**Figure 5 membranes-12-00361-f005:**
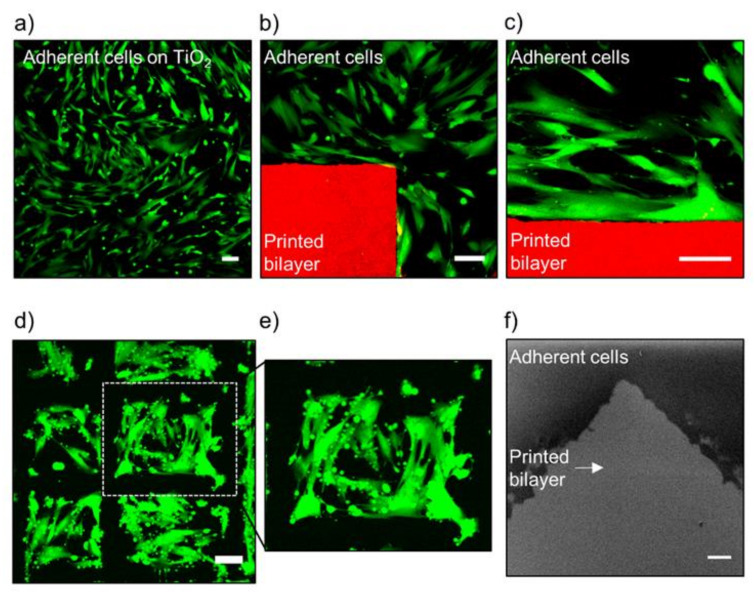
Adhesion of human fibroblast cells to a bare TiO_2_ surface with confinement by printed phospholipid bilayers: (**a**) Confocal image of adhered cells (green) on a bare TiO_2_ surface. (**b**) Confocal images of adhered cells (green) on a bare TiO_2_ surface with confinement by inkjet-printed phospholipid bilayers (red), and (**c**) its magnification. (**d**) Confinement of cell cultures (green) by printed lipid bilayers (not labeled, black), and (**e**) magnified view. (**f**) SEM characterization of the interface between a printed bilayer and cells. Confocal image scale bar: 100 µm, SEM image scale bar: 50 µm.

**Figure 6 membranes-12-00361-f006:**
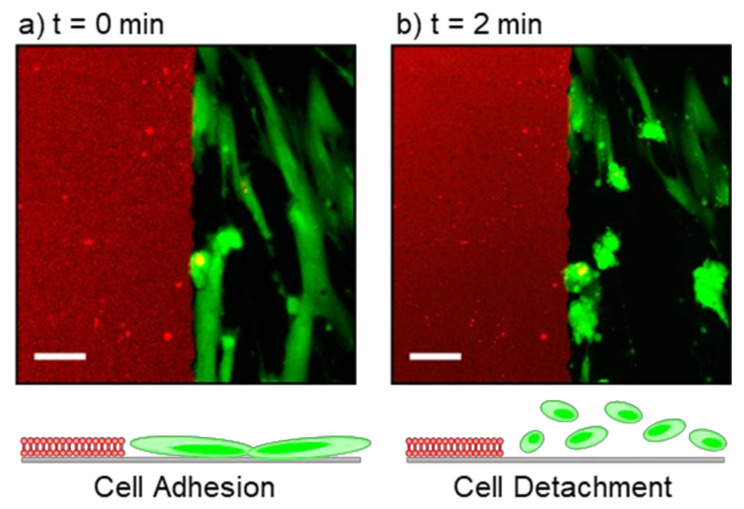
(**a**) Confocal images of adhered cells (green) alongside a printed tethered phospholipid bilayer (red) (t = 0 min). (**b**) Upon exposure to a confocal laser, cells gradually detached from the surface (t = 2 min). This effect was observed only in the presence of tethered phospholipid bilayers. Image scale bars: 50 µm.

## Data Availability

The data presented in this study are available upon reasonable request from the corresponding author.

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
