# Peer review of "Inkjet-Printed Phospholipid Bilayers on Titanium Oxide Surfaces: Towards Functional Membrane Biointerfaces"

_membranes, 2022, doi:10.3390/membranes12040361_

Round 1

Reviewer 1 Report

The manuscript by Cho et al. describes the effective combination of inkjet printing process with lipid surface chemistry for achieving the patterned DOCP lipid bilayers on titanium oxide surfaces. Interestingly, the process does not require any additional reagents and printed DOCP lipid bilayers displayed high durability, air and moisture stability with futuristic promising applications in titanium based implantable biomedical devices. The current protocol is interesting as it brings significant advancement to the area of functional biointerfaces. I believe that the work is of potential interest to the readership of Membranes and therefore, recommend publication followed by minor revisions as discussed below-

  1. In the Abstract, please highlight some important parameters related to the study like frequency and energy dissipation shifts, lateral mobility, diffusivity, etc. to describe the results.
  2. I suggest to include a Table comparing some previously reported inkjet-printed phospholipid bilayers with the current method to highlight the advantage of utilizing titanium oxide surfaces.
  3. Please increase the font size of legends shown Fig. 1c.
  4. Please cite some recent review papers (Materials 2016, 9(2), 116, Polymers 2022, 14(2), 238) related to biointerfaces interfaces in the Introduction Section.
  5. Line 74-75, please correct “lipid chemisorption to the surface” as “lipid chemisorption on the surface”.
  6. Line 76, please correct “hydrogen-bond breakers, and surfactant treatments, as well as exposure to albumin” as “hydrogen-bond breakers, surfactant treatments, as well as exposure to albumin”.
  7. Line 319-320, please rephrase the sentence “Notably, cells attached adjacent to the prints (Figures 3b,d), thus suggesting that the prints were not a disturbance to cell viability” as “Notably, the cells remain attached adjacent to the prints (Figures 3b,d)”, suggesting that the prints could not affect the cell viability”.

Author Response

Thank you for reviewing our manuscript and for offering excellent comments to help us improve the manuscript. Please see the attached file for properly formatted replies along with pasted replies below.

The manuscript by Cho et al. describes the effective combination of inkjet printing process with lipid surface chemistry for achieving the patterned DOCP lipid bilayers on titanium oxide surfaces. Interestingly, the process does not require any additional reagents and printed DOCP lipid bilayers displayed high durability, air and moisture stability with futuristic promising applications in titanium based implantable biomedical devices. The current protocol is interesting as it brings significant advancement to the area of functional biointerfaces. I believe that the work is of potential interest to the readership of Membranes and therefore, recommend publication followed by minor revisions as discussed below-

Reply: We thank the Reviewer for positive evaluation of our manuscript and for recommending publication after minor revision. Below, we have responded to each point individually and discuss how we have revised the manuscript accordingly.

  1. In the Abstract, please highlight some important parameters related to the study like frequency and energy dissipation shifts, lateral mobility, diffusivity, etc. to describe the results.

Reply: We thank the Reviewer for this excellent suggestion and have revised the abstract accordingly to include more technical specifics about the QCM-D and FRAP results as follows (additions in yellow highlight):

By combining advances in lipid surface chemistry and on-demand inkjet printing, we demonstrate the direct deposition and patterning of covalently tethered lipid bilayer membranes on titanium oxide surfaces, in ambient conditions and without any surface pretreatment process. The deposition conditions were evaluated by quartz crystal microbalance-dissipation (QCM-D) measurements, with corresponding resonance frequency (Δf) and energy dissipation (ΔD) shifts of around -25 Hz and <1 × 10-6, respectively, that indicated successful bilayer printing. The resulting printed phospholipid bilayers are stable in air and do not collapse following dehydration; through rehydration, the bilayers regain their functional properties such as lateral mobility (>1 µm2/s diffusion coefficient) according to fluorescence recovery after photobleaching (FRAP) measurements.

  1. I suggest to include a Table comparing some previously reported inkjet-printed phospholipid bilayers with the current method to highlight the advantage of utilizing titanium oxide surfaces.

Reply: We thank the Reviewer for this excellent suggestion. At present, there has only been one other inkjet-printed phospholipid bilayer paper reported in the literature and that was performed on a glass surface so it would be difficult to add a comprehensive table since the topic is newly emerging and there are currently few papers (just the previous one and our current one). Hence, to address this comment and a similar suggestion made by Reviewer 2, we have decided to add extended remarks in the conclusion of the revised manuscript to highlight the advantages and novelty of utilizing titanium oxide surfaces in terms of both potential application merits and technical performance specifics as follows (additions in yellow highlight):

“Compared to a previous report [20] that described inkjet printing of a phospholipid bilayer on a glass surface, there are several compelling merits and novel elements of the surface functionalization pursued here: (1) the current approach involves covalent attachment of the phospholipid bilayer to the TiO2 surface, whereas the previously printed bilayer on glass was only attached via noncovalent forces; and (2) no TiO2 surface pretreatment is required in the present case, whereas a polymerized lipid bilayer micropattern was first fabricated on the surface in the previous report prior to inkjet printing that resulted in a far more complex process overall. On the other hand, the streamlined fabrication capabilities we have developed in the present case, in turn enabled the printed DOCP lipid bilayers on TiO2 to exhibit high durability, as dehydration did not damage the printed patterns. Importantly, upon rehydration, the bilayer mobility was regained. This feature makes DOCP lipid bilayers highly useful for biomedical applications by overcoming typical sensitivity to air exposure and limitations of dehydration/rehydration.

  1. Please increase the font size of legends shown Fig. 1c.

Reply: In the revised manuscript, we have increased the font size of the legend in Fig 1c.

  1. Please cite some recent review papers (Materials 2016, 9(2), 116, Polymers 2022, 14(2), 238) related to biointerfaces interfaces in the Introduction Section.

Reply: We thank the Reviewer for this excellent suggestion and have added these two references to the Introduction.

  1. Line 74-75, please correct “lipid chemisorption to the surface” as “lipid chemisorption on the surface”.

Reply: We have revised the statement accordingly.

  1. Line 76, please correct “hydrogen-bond breakers, and surfactant treatments, as well as exposure to albumin” as “hydrogen-bond breakers, surfactant treatments, as well as exposure to albumin”.

Reply: We have revised statement accordingly, with slight modification to ensure grammatical correctness of the overall sentence.

  1. Line 319-320, please rephrase the sentence “Notably, cells attached adjacent to the prints (Figures 3b,d), thus suggesting that the prints were not a disturbance to cell viability” as “Notably, the cells remain attached adjacent to the prints (Figures 3b,d)”, suggesting that the prints could not affect the cell viability”.

Reply: We have revised the statement accordingly, with slight modification.

Reviewer 2 Report

This manuscript presents the direct deposition and patterning of covalently tethered lipid bilayer membranes on titanium oxide surfaces in ambient conditions for applications in patterned cell culture arrays. Without any surface pretreatment processes, the printed phospholipid bilayers are stable in air and exhibit stable functional properties such as lateral mobility after dehydration and rehydration.  The authors may want to address the following comments before consideration for publication.

  1. It would be helpful to compare the results from this study with the others reported in the literature to directly demonstrate the novelty.
  2. What is the smallest feature size that can be achieved in the printed lipid pattern on TiO2 substrates?  How about the large-scale capability and scalability of the proposed method?
  3. Does the result depend on the thickness of TiO2 substrates (with values relevant to those used in implantable devices)?
  4. Could the authors demonstrate the patterned cell culture arrays with a programmable pattern?
  5. It is helpful to elaborate on the discussion of the needs and applications of printing phospholipid bilayers on implantable devices with TiO2 coatings.

Author Response

Thank you for reviewing our manuscript and for offering excellent comments to help us improve the manuscript. Please see the attached file for properly formatted replies along with pasted replies below.

This manuscript presents the direct deposition and patterning of covalently tethered lipid bilayer membranes on titanium oxide surfaces in ambient conditions for applications in patterned cell culture arrays. Without any surface pretreatment processes, the printed phospholipid bilayers are stable in air and exhibit stable functional properties such as lateral mobility after dehydration and rehydration.  The authors may want to address the following comments before consideration for publication.

Reply: We thank the Reviewer for positive evaluation of our manuscript and for offering helpful suggestions to improve the manuscript. Below, we have responded to each point individually and discuss how we have revised the manuscript accordingly.

  1. It would be helpful to compare the results from this study with the others reported in the literature to directly demonstrate the novelty.

Reply: We agree with the Reviewer and inkjet printing of phospholipid bilayer coatings is indeed a newly emerging and highly novel fabrication strategy. Only one past paper in the literature reported the use of inkjet printing to fabricate noncovalently attached bilayer coatings on a glass surface and the inkjet-printing approach we took here is novel in terms of the lipid chemistry and extension to TiO2, which is a well-known challenging material on which to fabricate bilayer coatings. In the Conclusion section, we have added comparative discussion of these points and the compelling merits of the approach we took as follows (additions in yellow highlight):

“Compared to a previous report [20] that described inkjet printing of a phospholipid bilayer on a glass surface, there are several compelling merits and novel elements of the surface functionalization pursued here: (1) the current approach involves covalent attachment of the phospholipid bilayer to the TiO2 surface, whereas the previously printed bilayer on glass was only attached via noncovalent forces; and (2) no TiO2 surface pretreatment is required in the present case, whereas a polymerized lipid bilayer micropattern was first fabricated on the surface in the previous report prior to inkjet printing that resulted in a far more complex process overall. On the other hand, the streamlined fabrication capabilities we have developed in the present case, in turn enabled the printed DOCP lipid bilayers on TiO2 to exhibit high durability, as dehydration did not damage the printed patterns. Importantly, upon rehydration, the bilayer mobility was regained. This feature makes DOCP lipid bilayers highly useful for biomedical applications by overcoming typical sensitivity to air exposure and limitations of dehydration/rehydration.

  1. What is the smallest feature size that can be achieved in the printed lipid pattern on TiO2 substrates?  How about the large-scale capability and scalability of the proposed method?

Reply: Based on the optimized printing conditions that we developed in this study, the smallest feature size that we could achieve was ~40 µm and the inkjet-printing approach is able to fabricate large-scale areas. In the revised manuscript, we added the following remarks to cover these points (additions in yellow highlight):

“Pattern features as small as 40 µm were achieved based on the optimized printer settings used in this case, and the inkjet printer from this study can be potentially expanded to areas as large as 500 mm × 500 mm, especially if future design iterations can bypass the need for preparing lipid vesicles as new coating methods emerge [56-58].

  1. Does the result depend on the thickness of TiO2 substrates (with values relevant to those used in implantable devices)?

Reply: We thank the Reviewer for this great question. The result does not depend on the thickness of the TiO2 substrate provided there is a minimum coating thickness of ~2 nm. In our experiments, the coats were in the thickness range of 15-50 nm, while the oxide film thickness on titanium-based implants can typically vary in the range of 3-1000 nm. In the revised manuscript, we have added extended remarks concerning this point as follows (additions in yellow highlight):

“From an applied perspective, the combination of DOCP lipid bilayer durability on TiO2 surfaces and convenient inkjet printing deposition increases the applicability of the planar bilayer as a bioinspired coating technology, which could be advantageous for titanium-based medical implants that typically have TiO2 oxide film thicknesses ranging from around 3 to 1000 nm [59]. In the present case, the film thickness was ~15 to ~50 nm in the different experiments but the fabrication approach is anticipated to be broadly useful across a wide range of film thickness values because the chemisorption process only depends on coordinate bond formation between DOCP lipid molecules and the uppermost TiO2 surface layer.

  1. Could the authors demonstrate the patterned cell culture arrays with a programmable pattern?

Reply: We thank the Reviewer for this excellent suggestion. In this study, we used a relatively simple, square-type pattern while more sophisticated patterns are also possible and could be advantageous to modulate cell-surface interactions. In the revised manuscript, we have added the following remarks to cover these design possibilities in future studies (additions in yellow highlight):

“While the pattern design we used was relatively simple in this proof-of-concept study (cf. Figure 5d), more sophisticated patterns could potentially be incorporated to modulate cell-implant interactions by controlling pattern shape and dimensions.

  1. It is helpful to elaborate on the discussion of the needs and applications of printing phospholipid bilayers on implantable devices with TiO2 coatings.

Reply: We thank the Reviewer for this excellent suggestion and it is a great point. In practice, both full-spanning and patterned bilayer coatings could be potentially useful and the latter option is particularly intriguing in terms of modulating cell-surface interactions. To discuss these possibilities, in the revised manuscript, we have added the following remarks in the revised manuscript (additions in yellow highlight):

“It might be advantageous to use the patterned bilayer coatings to corral cells to interact with implant surfaces in a more controlled manner while modulating cell-cell interactions and potentially improving biocompatibility. Such functionality might be utilized to prevent unwanted cell growth on implants and medical devices at strategic locations and the bilayer coating itself could also reduce nonspecific adsorption of immune-related proteins.

Round 2

Reviewer 2 Report

comments addressed